# Securing Secure Aggregation: Mitigating Multi-Round Privacy Leakage in Federated Learning

## Abstract

Secure aggregation is a critical component in federated learning (FL), which enables the server to learn the aggregate model of the users without observing their local models. Conventionally, secure aggregation algorithms focus only on ensuring the privacy of individual users in a *single* training round. We contend that such designs can lead to significant privacy leakages over *multiple* training rounds, due to partial user selection/participation at each round of FL. In fact, we show that the conventional random user selection strategies in FL may lead to leaking users' individual models within a number of rounds that is linear in the number of users. To address this challenge, we introduce a secure aggregation framework, Multi-RoundSecAgg, with multi-round privacy guarantees. In particular, we introduce a new metric to quantify the privacy guarantees of FL over multiple training rounds, and develop a structured user selection strategy that guarantees the long-term privacy of each user (over any number of training rounds). Our framework also carefully accounts for the fairness and the average number of participating users at each round. Our experiments on MNIST, CIFAR-10 and CIFAR-100 datasets in the IID and the non-IID settings demonstrate the performance improvement over the baselines, both in terms of privacy protection and test accuracy.

## 1 Introduction

Federated learning (FL) enables collaborative training of machine learning models over the data collected and stored locally by multiple data-owners. The training in FL is typically coordinated by a central server who maintains a global model that is updated locally by the users. The local updates are then aggregated by the server to update the global model. Throughout the training process, the users never share their data with the server, i.e., the data is always kept on device, rather, they only share their local updates. However, as has been shown recently, the local models may still reveal substantial information about the local datasets, and the private training data can be reconstructed from the local models through inference or inversion attacks (see e.g., [11, 26, 42, 12]).

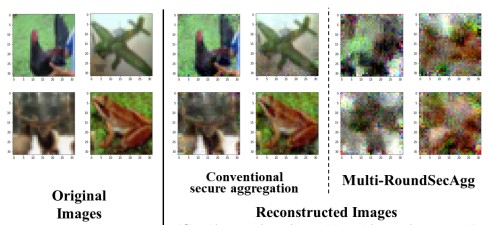

**Original Images**    **Conventional secure aggregation**    **Multi-RoundSecAgg**

**Reconstructed Images (Gradient estimation + Model inversion attack)**

**Figure 1:** A qualitative comparison of the reconstructed images in two settings is shown. The first setting corresponds to the case that model privacy with random user selection (e.g., `FedAvg` [25]) is protected by conventional secure aggregation schemes as [4] at each round. In the second setting, our proposed method ensures the long-term privacy of individual models over any number of rounds, and hence model inversion attack cannot work well. This reconstruction process is described in detail in Appendix H.

To prevent such information leakage, *secure aggregation* protocols are proposed (e.g., [4, 31, 15, 40, 2, 38, 30]) to protect the privacy of the local models, both from the server and the other users, while still allowing the server to learn their aggregate. More specifically, the secure aggregation protocols ensure that, at any given round, the server can only learn the aggregate model of the users, and beyond that no further information is revealed about the individual model.

Secure aggregation protocols, however, only ensure the privacy of the individual users in a *single training round*, and do not consider their privacy over multiple training rounds [4, 2, 31, 32]. On the other hand, due to partial user selection [7, 5, 6, 28], the server may be able to reconstruct the individual models of some users using the aggregated models from the previous rounds. In fact, we show that after a sufficient number of rounds, all local models can be recovered with a high accuracy if the server uniformly chooses a random subset of the users to participate at every round. As shown

in Fig.1, performing model inversion attack [12] with the recovered local models yields reconstructed images with a similar quality as the original images.

**Contributions**. As such motivated, we study long-term user privacy in FL. Specifically, our contributions are as follows.

1. We introduce a new metric to capture long-term privacy guarantees for secure aggregation protocols in FL for the first time. This long-term privacy requires that the server cannot reconstruct any individual model using the aggregated models from any number of training rounds. Using this metric, we show that the conventional random selection schemes can result in leaking the local models after a sufficient number of rounds, even if secure aggregation is employed at each round.
2. We propose Multi-RoundSecAgg, a privacy-preserving structured user selection strategy that ensures the long-term privacy of the individual users over any number of training rounds. This strategy also takes into account the fairness of the selection process and the average number of participating users at each round.
3. We demonstrate that Multi-RoundSecAgg creates a trade-off between the long-term privacy guarantee and the average number of participating users. In particular, as the average number of participating users increases, the long-term privacy guarantee becomes weaker.
4. We provide the convergence analysis of Multi-RoundSecAgg, which shows that the long-term privacy guarantee and the average number of participating users control the convergence rate. The convergence rate is maximized when the average number of participating users is maximized. (e.g., the random user selection strategy maximizes the average number of participating users at the expense of not providing long-term privacy guarantees). As we require stronger long-term privacy guarantees, the average number of participating users decreases and a larger number of training rounds is required to achieve the same level of accuracy as the random selection strategy.
5. Finally, our experiments in both IID and non-IID settings on MNIST, CIFAR-10 and CIFAR-100 datasets demonstrate that Multi-RoundSecAgg achieves almost the same test accuracy compared to the random selection scheme while providing better long-term privacy guarantees.

## 2   Related Work

The underlying principle of the secure aggregation protocol in [4] is that each pair of users exchange a pairwise secret key which they can use to mask their local models before sharing them with the server. The pairwise masks cancel out when the server aggregates the masked models, allowing the server to aggregate the local models. These masks also ensure that the local models are kept private, i.e., no further information is revealed beyond the aggregate of the local models. This protocol, however, incurs a significant communication cost due to exchanging and reconstructing the pairwise keys.

Recently, several works have developed computation and communication-efficient protocols [31, 15, 2, 35, 8, 10, 38], which are complementary to and can be combined with our work. Another line of work focused on designing partial user selection strategies to overcome the communication bottleneck in FL while speeding up the convergence by selecting the users based on their local loss [7, 5, 6, 28].

Previous works, either on secure aggregation or on partial user selection, however, do not consider mitigating the potential privacy leakage as a result of partial user participation and the server observing the aggregated models across multiple training rounds. While [27] pointed out to the privacy leakage of secure aggregation, mitigating this leakage has not been considered and our work is the first secure aggregation protocol to address this challenge.

Differential privacy (DP), in which each user adds artificial noises to the local models, can be one of the potential solution to protect the privacy leakage over the multiple rounds [9, 1, 37, 3, 16]. In DP, however, the privacy guarantee sacrifices the model performance, which is known as a privacy-utility trade-off. It is worth noting that secure aggregation and DP are complementary, i.e., all the benefits of DP can be applied to our approach by adding noise to the local models [3]. In this paper, our objective is to understand the secure aggregation itself.

## 3   System Model

In this section, we first describe the basic federated learning model in Section 3.1. Next, we introduce the multi-round secure aggregation problem for federated learning and define the key metrics to evaluate the performance of a multi-round secure aggregation protocol in Section 3.2.

## 3.1 Basic Federated Learning Model

We consider a cross-device federated learning setup consisting of a server and $N$ users. User $i \in [N]$ has a local dataset $\mathcal{D}_i$ consisting of $m_i = |\mathcal{D}_i|$ data samples. The users are connected to each other through the server, i.e., all communications between the users goes through the server [24, 4, 17]. The goal is to collaboratively learn a global model $x$ with dimension $d$, using the local datasets that are generated, stored, and processed locally by the users. The training task can be represented by minimizing a global loss function,

$$\min_x L(x) \text{ s.t. } L(x) = \frac{1}{\sum_{i=1}^N w_i} \sum_{i=1}^N w_i L_i(x), \tag{1}$$

where $L_i$ is the loss function of user $i$ and $w_i \geq 0$ is a weight parameter assigned to user $i$ to specify the relative impact of that user. A common choice for the weight parameters is $w_i = m_i$ [17]. We define the optimal model parameters $x^*$ and $x_i^*$ as $x^* = \arg\min_{x \in \mathbb{R}^d} L(x)$ and $x_i^* = \arg\min_{x \in \mathbb{R}^d} L_i(x)$.

**Federated Averaging with Partial User Participation.** To solve (1), the most common algorithm is the *FedAvg* (federated averaging) algorithm [24]. *FedAvg* is an iterative algorithm, where the model training is done by repeatedly iterating over individual local updates. At the beginning of training round $t$, the server sends the current state of the global model, denoted by $x^{(t)}$, to the users. Each round consists of two phases, local training and aggregation. In the local training phase, user $i \in [N]$ updates the global model by carrying out $E$ ($\geq 1$) local stochastic gradient descent (SGD) steps and sends the updated local model $x_i^{(t)}$ to the server. One of key features of cross-device FL is partial device participation. Due to various reasons such as unreliable wireless connectivity, or battery issues, at any given round, only a fraction of the users are available to participate in the protocol. We refer to such users as *available* users throughout the paper. In the aggregation phase, the server selects $K \leq N$ users among the available users if this is possible and aggregates their local updates. The server updates the global model as follows

$$x^{(t+1)} = \sum_{i \in \mathcal{S}^{(t)}} w_i' x_i^{(t)} = \mathbf{X}^{(t)\top} p^{(t)}, \tag{2}$$

where $\mathcal{S}^{(t)}$ is the set of participating users at round $t$, $w_i' = \frac{w_i}{\sum_{i \in \mathcal{S}^{(t)}} w_i}$, and $p^{(t)} \in \{0, 1\}^N$ is the corresponding characteristic vector. That is, $p^{(t)}$ denotes a participation vector at round $t$ whose $i$-th entry is 0 when user $i$ is not selected and 1 otherwise. $\mathbf{X}^{(t)}$ denotes the concatenation of the weighted local models at round $t$, i.e., $\mathbf{X}^{(t)} = \left[w_1' x_1^{(t)}, \ldots, w_N' x_N^{(t)}\right]^\top \in \mathbb{R}^{N \times d}$. Finally, the server broadcasts the updated global model $x^{(t+1)}$ to the users for the next round.

**Threat Model.** Similar to the prior works on secure aggregation as [4, 15, 31], we consider the honest-but-curious model. All participants follow the protocol honestly in this model, but try to learn as much as possible about the users. At each round, the privacy of individual model $x_i^{(t)}$ in (2) is protected by secure aggregation such that the server only learns the aggregated model $\sum_{i \in \mathcal{S}^{(t)}} w_i' x_i^{(t)}$.

## 3.2 Multi-round Secure Aggregation

Conventional secure aggregation protocols only consider the privacy guarantees over a single training round. While secure aggregation protocols have provable privacy guarantees at any single round, in the sense that no information is leaked beyond the aggregate model at each round, the privacy guarantees do not extend to attacks *that span multiple training rounds*. Specifically, by using the aggregate models and participation information across multiple rounds, an individual model may be reconstructed. For instance, consider the following user participation strategy across three training rounds, $p^{(1)} = [1, 1, 0]^\top$, $p^{(2)} = [0, 1, 1]^\top$, and $p^{(3)} = [1, 0, 1]^\top$. Assume a scenario where the local updates do not change significantly over time (e.g., models start to converge, or the server fixes the global model over consecutive rounds), i.e., $x_i = x_i^{(t)}$ for all $i \in [3]$ and $t \in [3]$. Then, the server can single out individual model, e.g., $x_1 = (x^{(1)} + x^{(3)} - x^{(2)})/2$. Similarly, the server can single out all individual models $x_i$, even if a secure aggregation protocol is employed at each round.

In this paper, we study secure aggregation protocols with long-term privacy guarantees (which we term *multi-round secure aggregation*) for the cross-device FL setup which has not been studied before.

We assume that user $i \in [N]$ drops from the protocol at each round with probability $p_i$. $\mathcal{U}^{(t)}$ denotes the index set of available users at round $t$ and $\boldsymbol{u}^{(t)} \in \{0, 1\}^N$ is a vector indicating the available users such that $\{\boldsymbol{u}^{(t)}\}_j = \mathbb{1}\{j \in \mathcal{U}^{(t)}\}$, where $\{\boldsymbol{u}\}_j$ is $j$-th entry of $\boldsymbol{u}$ and $\mathbb{1}\{\cdot\}$ is the indicator function. The server selects $K$ users from $\mathcal{U}^{(t)}$, if $|\mathcal{U}^{(t)}| \geq K$, based on the history of selected users in previous rounds. If $|\mathcal{U}^{(t)}| < K$, the server skips this round. The local models of the selected users are then aggregated via a secure aggregation protocol (i.e., by communicating masked models), at the end of which the server learns the aggregate of the local models of the selected users. Our goal is to design a user selection algorithm $\mathcal{A}^{(t)} : \{0, 1\}^{t \times N} \times \{0, 1\}^N \to \{0, 1\}^N$,

$$\mathcal{A}^{(t)}\big(\mathbf{P}^{(t)}, \boldsymbol{u}^{(t)}\big) = \boldsymbol{p}^{(t)} \text{ such that } \|\boldsymbol{p}^{(t)}\|_0 \in \{0, K\}, \tag{3}$$

to prevent the potential information leakage over multiple rounds, where $\boldsymbol{p}^{(t)} \in \{0, 1\}^N$ is the participation vector defined in (2), $\|\boldsymbol{x}\|_0$ denotes the $L_0$-"norm" of a vector $\boldsymbol{x}$ and $K$ denotes the number of selected users. We note that $\mathcal{A}^{(t)}$ can be a random function. $\mathbf{P}^{(t)}$ is a matrix representing the user participation information up to round $t$, and is termed the *participation matrix*, given by

$$\mathbf{P}^{(t)} = \big[\boldsymbol{p}^{(0)}, \boldsymbol{p}^{(1)}, \ldots, \boldsymbol{p}^{(t-1)}\big]^{\top} \in \{0, 1\}^{t \times N}. \tag{4}$$

**Key Metrics.** A multi-round secure aggregation protocol can be represented by $\mathcal{A} = \{\mathcal{A}^{(t)}\}_{t \in [J]}$, where $\mathcal{A}^{(t)}$ is the user selection algorithm at round $t$ defined in (3) and $J$ is the total number of rounds. The inputs of $\mathcal{A}^{(t)}$ are a random vector $\boldsymbol{u}^{(t)}$, which indicates the available users at round $t$, and the participation matrix $\mathbf{P}^{(t)}$ defined in (4) which can be a random matrix. Given the participation matrix $\mathbf{P}^{(J)}$, we evaluate the performance of the corresponding multi-round secure aggregation protocol through the following metrics.

1. **Multi-round Privacy Guarantee.** The secure aggregation protocols ensure that the server can only learn the sum of the local models of some users in each single round, but they do not consider what the server can learn over the long run. Our multi-round privacy definition extends the guarantees of the secure aggregation protocols from one round to all rounds by requiring that the server can only learn a sum of the local models even if the server exploits the aggregate models of all rounds. That is, our multi-round privacy guarantee is a natural extension of the privacy guarantee provided by the secure aggregation protocols considering a single training round. Specifically, a multi-round privacy guarantee $T$ requires that any non-zero partial sum of the local models that the server can reconstruct, through any linear combination $\mathbf{X}^{\top}\mathbf{P}^{(J)^{\top}}\boldsymbol{z}$, where $\boldsymbol{z} \in \mathbb{R}^J \setminus \{\mathbf{0}\}$, must be of the form[1]

$$\mathbf{X}^{\top}\mathbf{P}^{(J)^{\top}}\boldsymbol{z} = \sum_{i \in [n]} a_i \sum_{j \in \mathcal{S}_i} \boldsymbol{x}_j = a_1 \sum_{j \in \mathcal{S}_1} \boldsymbol{x}_j + a_2 \sum_{j \in \mathcal{S}_2} \boldsymbol{x}_j + \cdots + a_n \sum_{j \in \mathcal{S}_n} \boldsymbol{x}_j, \tag{5}$$

where $|\mathcal{S}_i| \geq T, a_i \neq 0, \forall i \in [n]$ and $n \in \mathbb{Z}^+$. Here all the sets $\mathcal{S}_i$, the number of sets $n$, and each $a_i$ could all depend on $\boldsymbol{z}$. In equation (5), we consider the worst-case scenario, where the local models do not change over the rounds. That is, $\mathbf{X}^{(t)} = \mathbf{X}, \forall t \in [J]$. Intuitively, this guarantee ensures that the best that the server can do is to reconstruct a partial sum of $T$ local models which corresponds to the case where $n = 1$. When $T \geq 2$, this condition implies that the server cannot get any user model from the aggregate models of all training rounds (the best it can obtain is the sum of two local models).

**Remark 1.** (Weaker Privacy Notion). It is worth noting that, a weaker privacy notion would require that $\|\mathbf{P}^{(J)^{\top}}\boldsymbol{z}\|_0 \geq T$ when $\mathbf{P}^{(J)^{\top}}\boldsymbol{z} \neq \mathbf{0}$. When $T = 2$, this definition requires that the server cannot reconstruct any individual model (the best it can do is to obtain a linear combination of two local models). This notion, however, allows constructions in the form of $a\boldsymbol{x}_i + b\boldsymbol{x}_j$ for any $a, b \in \mathbb{R} \setminus \{0\}$. When $a \gg b$, however, this is almost the same as recovering $\boldsymbol{x}_i$ perfectly, hence this privacy criterion is weaker than that of (5).

**Remark 2.** (Multi-round Privacy of Random Selection). In Section 6, we empirically show that a random selection strategy in which $K$ available users are selected uniformly at random at each round does not ensure multi-round privacy even with respect to the weaker definition of Remark 1. Specifically, the local models can be reconstructed within a number of rounds that is linear in $N$. We also show theoretically in Appendix H that when $\min(N - K, K) \geq cN$, where $c > 0$ is a constant, then the probability that the server can reconstruct all local models after $N$ rounds is

---

[1]We assume that $w_i = \frac{1}{N}, \forall i \in [N]$ in this paper.

at least $1 - 2e^{-c'N}$ for a constant $c'$ that depends on $c$. Finally, we show that a random selection scheme in which the users are selected in an i.i.d fashion according to $\text{Bern}(\frac{K}{N(1-p)})$ reveals all local models after $N$ rounds with probability that converges to 1 exponentially fast.

**Remark 3.** (Worst-Case Assumption). In (5), we considered the worst-case assumption where the models do not change over time. When the local models change over rounds, the multi-round privacy guarantee becomes even stronger as the number of unknowns increases. In Fig. 1 and Appendix H, we empirically show that the conventional secure aggregation schemes leak extensive information of training data even in the realistic settings where the models change over the rounds.

2. **Aggregation Fairness Gap.** The average aggregation fairness gap quantifies the largest gap between any two users in terms of the expected relative number of rounds each user has participated in training. Formally, the average aggregation fairness gap is defined as follows

$$F = \max_{i \in [N]} \limsup_{J \to \infty} \frac{1}{J} \mathbb{E}\Big[ \sum_{t=0}^{J-1} \mathbb{1}\big\{ \{\boldsymbol{p}^{(t)}\}_i = 1 \big\} \Big] - \min_{i \in [N]} \liminf_{J \to \infty} \frac{1}{J} \mathbb{E}\Big[ \sum_{t=0}^{J-1} \mathbb{1}\big\{ \{\boldsymbol{p}^{(t)}\}_i = 1 \big\} \Big], \quad (6)$$

where $\{\boldsymbol{p}^{(t)}\}_i$ is $i$-th entry of the vector $\boldsymbol{p}^{(t)}$ and the expectation is over the randomness of the user selection algorithm $\mathcal{A}$ and the user availability. The main intuition behind this definition is that when $F = 0$, all users participate on average on the same number of rounds. This is important to take the different users into consideration equally and our experiments show that the accuracy of the schemes with small $F$ are much higher than the schemes with high $F$.

3. **Average Aggregation Cardinality.** The aggregation cardinality quantifies the expected number of models to be aggregated per round. Formally, it is defined as

$$C = \liminf_{J \to \infty} \frac{\mathbb{E}\big[ \sum_{t=0}^{J-1} \|\boldsymbol{p}^{(t)}\|_0 \big]}{J}, \quad (7)$$

where the expectation is over the randomness in $\mathcal{A}$ and the user availability. Intuitively, less number of rounds are needed to converge as more users participate in the training. In fact, as we show in Section 5.2, $C$ directly controls the convergence rate.

### 3.3 Baseline Schemes

In this subsection, we introduce three baseline schemes for multi-round secure aggregation.

**Random Selection.** In this scheme, at each round, the server selects $K$ users at random from the set of available users if this is possible.

**Random Weighted Selection.** This scheme is a modified version of random selection to reduce $F$ when the dropout probabilities of the users are not equal. Specifically, $K$ users are selected at random from the available users with the minimum frequency of participation in the previous rounds.

**User Partitioning (Grouping).** In this scheme, the users are partitioned into $G = N/K$ equal-sized groups denoted as $\mathcal{G}_1, \mathcal{G}_2, \cdots, \mathcal{G}_G$. At each round, the server selects one of the groups if none of the users in this group has dropped out. If multiple groups are available, to reduce the aggregation fairness gap, the server selects a group including a user with the minimum frequency of participation in previous rounds. If no group is available, the server skips this round.

## 4 Proposed Scheme: Multi-RoundSecAgg

In this section, we present Multi-RoundSecAgg, which has two components as follows.

- The first component designs a family of sets of users that satisfy the multi-round privacy requirement. The inputs of the first component are the number of users ($N$), the number of selected users at each round ($K$), and the desired multi-round privacy guarantee ($T$). The output is a family of sets of $K$ users satisfying the multi-round privacy guarantee $T$, termed as a *privacy-preserving family*. This family is represented by a matrix **B**, where the rows are the characteristic vectors of these user sets.

- The second component selects a set from this designed family to satisfy the fairness guarantee. The inputs to the second component are the family **B**, the set of available users at round $t$, $\mathcal{U}^{(t)}$, and the frequency of participation of each user. The output is the set of users that will participate at round $t$.

We now describe these two components in detail.

**Component 1 (Batch Partitioning (BP) of the users to guarantee multi-round privacy).** The first component designs a family of $R_{\text{BP}}$ sets, where $R_{\text{BP}}$ is the size of the set, satisfying the multi-round privacy requirement $T$. We denote the $R_{\text{BP}} \times N$ binary matrix corresponding to these sets by $\mathbf{B} = [\boldsymbol{b}_1, \cdots, \boldsymbol{b}_{R_{\text{BP}}}]^\top$, where $\|\boldsymbol{b}_i\|_0 = K, \forall i \in [R_{\text{BP}}]$. That is, the rows of $\mathbf{B}$ are the characteristic vectors of those sets. The main idea of our scheme is to restrict certain sets of users of size $T$, denoted as batches, to either participate together or not participate at all. This guarantees a multi-round privacy $T$ as we show in Section 5.

To construct a family of sets with this property, the users are first partitioned into $N/T$ batches. At any given round, either all or none of the users of a particular batch participate in training. The server can choose $K/T$ batches to participate in training, provided that all users in any given selected batch are available. Since there are $\binom{N/T}{K/T}$ possible sets with this property, then the size of this privacy-preserving family of sets is given by $R_{\text{BP}} \stackrel{\text{def}}{=} \binom{N/T}{K/T}^2$.

In the extreme case of $T = 1$, this strategy specializes to random selection where the server can choose any $K$ possible users. In the other extreme case of $T = K$, this strategy specializes to the partitioning strategy where there are $N/K$ possible sets. We next provide an example to illustrate the construction of $\mathbf{B}$.

$$\mathbf{B} = \begin{bmatrix} 1 & 1 & 1 & 1 & 0 & 0 & 0 & 0 \\ 1 & 1 & 0 & 0 & 1 & 1 & 0 & 0 \\ 1 & 1 & 0 & 0 & 0 & 0 & 1 & 1 \\ 0 & 0 & 1 & 1 & 1 & 1 & 0 & 0 \\ 0 & 0 & 1 & 1 & 0 & 0 & 1 & 1 \\ 0 & 0 & 0 & 0 & 1 & 1 & 1 & 1 \end{bmatrix}$$

**Figure 2:** Example of our construction with $N = 8, K = 4$ and $T = 2$.

**Example 1** ($N = 8, K = 4, T = 2$). *In this example, the users are partitioned into 4 batches as $\mathcal{G}_1 = \{1, 2\}, \mathcal{G}_2 = \{3, 4\}, \mathcal{G}_3 = \{5, 6\}$ and $\mathcal{G}_4 = \{7, 8\}$ as given in Fig. 2. The server can choose any two batches out of these 4 batches, hence we have $R_{BP} = \binom{4}{2} = 6$ possible sets. This ensures a multi-round privacy $T = 2$.*

**Component 2 (Available batch selection to guarantee fairness).** At round $t$, user $i \in [N]$ is available to participate in the protocol with a probability $1 - p_i \in (0, 1]$. The frequency of participation of user $i$ before round $t$ is denoted by $f_i^{(t)} \stackrel{\text{def}}{=} \sum_{j=0}^{t-1} \mathbb{1}\left\{\{\boldsymbol{p}^{(j)}\}_i = 1\right\}$. Given the set of available users at round $t$, $\mathcal{U}^{(t)}$, and the frequencies of participation $\boldsymbol{f}^{(t-1)} = (f_1^{(t-1)}, \cdots, f_N^{(t-1)})$, the server selects $K$ users. To do so, the server first finds the submatrix of $\mathbf{B}$ denoted by $\mathbf{B}^{(t)}$ corresponding to $\mathcal{U}^{(t)}$. Specifically, the $i$-th row of $\mathbf{B}$ denoted by $\boldsymbol{b}_i^\top$ is included in $\mathbf{B}^{(t)}$ provided that $\text{supp}(\boldsymbol{b}_i) \subseteq \mathcal{U}^{(t)}$. If $\mathbf{B}^{(t)}$ is an empty matrix, then the server skips this round. Otherwise, the server selects a row from $\mathbf{B}^{(t)}$ uniformly at random if $p_i = p, \forall i \in [N]$. If the users have different $p_i$, the server selects a row from $\mathbf{B}^{(t)}$ that includes the user with the minimum frequency of participation $\ell_{\min}^{(t-1)} \stackrel{\text{def}}{=} \arg\min_{i \in \mathcal{U}^{(t)}} f_i^{(t-1)}$. If there are many such rows, then the server selects one of them uniformly at random.

**Remark 4.** (Necessity of the Second Component). The second component is necessary to guarantee that the aggregation fairness gap goes to zero as we show in Theorem 1 and Section 6.

Overall, the algorithm first designs a privacy-preserving family of sets to ensure the multi-round privacy guarantee $T$. Then specific sets are selected from this family to ensure fairness. We describe the two components of Multi-RoundSecAgg in detail in Algorithm 1 and Algorithm 2 in Appendix D.

## 5 Theoretical Results

In this section, we provide the theoretical guarantees of Multi-RoundSecAgg in Section 5.1 and the convergence analysis of Multi-RoundSecAgg in Section 5.2.

### 5.1 Theoretical Guarantees of Multi-RoundSecAgg

In this subsection, we establish the theoretical guarantees of Multi-RoundSecAgg in terms of the multi-round privacy guarantee, the aggregation fairness gap and the average aggregation cardinality.

**Theorem 1.** *Multi-RoundSecAgg with parameters $N, K, T$ ensures a multi-round privacy guarantee of $T$, an aggregation fairness gap $F = 0$, and an average aggregation cardinality given by*

$$C = K\left(1 - \sum_{i=N/T-K/T+1}^{N/T} \binom{N/T}{i} q^i (1-q)^{N/T-i}\right),$$

---

[2] We assume for simplicity that $N/T$ and $K/T$ are integers.

where $q = 1 - (1 - p)^T$, when all users have the dropout probability $p$.

We provide the proof of Theorem 1 in Appendix A.

**Remark 5.** (Trade-off between "Multi-round Privacy Guarantee" and "Average Aggregation Cardinality"). Theorem 1 indicates a trade-off between the multi-round privacy and the average aggregation cardinality since as $T$ increases, $C$ decreases which slows down the convergence as we show in Sec. 5.2. We show this trade-off in Fig. 3.

**Remark 6.** (Necessity of Batch Partitioning (BP)). We show that any strategy that satisfies the privacy guarantee in Equation (5) must have a batch partitioning structure, and for given $N, K, T, K \leq N/2$, the largest number of distinct user sets in any strategy is at most $\binom{N/T}{K/T}$, which is achieved in our design in Section 4. We provide the proof in Appendix C.

**Remark 7.** (Non-linear Reconstructions of Aggregated Models). The privacy criterion in Eq. (5) considers linear reconstructions of the aggregated models. One may also consider more general non-linear reconstructions. The long-term privacy guarantees of batch partitioning hold even under such reconstructions as the users in the same batch always participate together or do not participate at all. Hence, the server cannot separate individual models within the same batch even through non-linear operations.

## 5.2 Convergence Analysis of Multi-RoundSecAgg

For convergence analysis of Multi-RoundSecAgg, we first introduce a few common assumptions [23, 39].

**Assumption 1.** $L_1, \ldots, L_N$ in (1) are all $\rho$-smooth: for all $a, b \in \mathbb{R}^d$ and $i \in [N]$, $L_i(a) \leq L_i(b) + (a - b)^\top \nabla L_i(b) + \frac{\rho}{2} \|a - b\|^2$.

**Assumption 2.** $L_1, \ldots, L_N$ in (1) are all $\mu$-strongly convex: for all $a, b \in \mathbb{R}^d$ and $i \in [N]$, $L_i(a) \geq L_i(b) + (a - b)^\top \nabla L_i(b) + \frac{\mu}{2} \|a - b\|^2$.

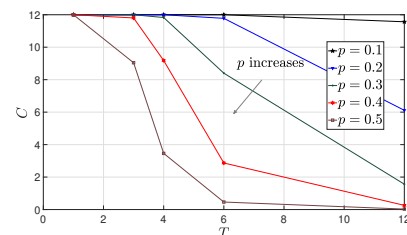

**Figure 3:** An illustration of the trade-off between the multi-round privacy guarantee $T$ and the average aggregation cardinality $C$. In this example, $N = 120$ and $K = 12$.

**Assumption 3.** Let $\xi_i^{(t)}$ be a sample uniformly selected from the dataset $\mathcal{D}_i$. The variance of the stochastic gradients at each user is bounded, i.e., $\mathbb{E}\|\nabla L_i(x_i^{(t)}, \xi_i^{(t)}) - \nabla L_i(x_i^{(t)})\|^2 \leq \sigma_i^2$ for $i \in [N]$.

**Assumption 4.** The expected squared norm of the stochastic gradients is uniformly bounded, i.e., $\mathbb{E}\|\nabla L_i(x_i^{(t)}, \xi_i^{(t)})\|^2 \leq G^2$ for all $i \in [N]$.

We now state the convergence guarantees of Multi-RoundSecAgg.

**Theorem 2.** *Consider a FL setup with $N$ users to train a machine learning model from (1). Assume $K$ users are selected by Multi-RoundSecAgg with average aggregation cardinality $C$ defined in (7) to update the global model from (2), and all users have the same dropout rate, hence Multi-RoundSecAgg selects a random set of $K$ users uniformly from the set of available user sets at each round. Then, the following is satisfied*

$$\mathbb{E}[L(x^{(J)})] - L^* \leq \frac{\rho}{\gamma + \frac{C}{K} EJ - 1} \left( \frac{2(\alpha + \beta)}{\mu^2} + \frac{\gamma}{2} \mathbb{E}\|x^{(0)} - x^*\|^2 \right), \tag{8}$$

*where $\alpha = \frac{1}{N} \sum_{i=1}^N \sigma_i^2 + 6\rho\Gamma + 8(E-1)^2 G^2$, $\beta = \frac{4(N-K)E^2 G^2}{K(N-1)}$, $\Gamma = L^* - \sum_{i=1}^N L_i^*$, and $\gamma = \max\left\{ \frac{8\rho}{\mu}, E \right\}$.*

We provide the proof of Theorem 2 in Appendix B.

**Remark 8.** (The average aggregation cardinality controls the convergence rate.) Theorem 2 shows how the average aggregation cardinality affects the convergence. When the average aggregation cardinality is maximized, i.e., $C = K$, the convergence rate in Theorem 2 equals that of the random selection algorithm provided in Theorem 3 of [23]. In (8), we have the additional term $E$ (number of local epochs) in front of $J$ compared to Theorem 3 of [23] as we use global round index $t$ instead of using step index of local SGD. As the average aggregation cardinality decreases, a greater number of training rounds is required to achieve the same level of accuracy.

**Remark 9.** (General Convex and Non-Convex Convergence Rates). Theorem 2 considers the strongly-convex case, but we consider the general convex and the non-convex cases in Appendix I.

**Remark 10.** (Different Dropout Rates). When the dropout probabilities of the users are not the same, characterizing the convergence guarantees of Multi-RoundSecAgg is challenging. This is due to the fact that batch selection based on the frequency of participation breaks the conditional unbiasedness of the user selection, which is required for the convergence guarantee. In experiments, however, we empirically show that Multi-RoundSecAgg guarantees the convergence with different dropout rates.

# 6 Experiments

Our experiments consist of two parts. We first numerically demonstrate the performance of Multi-RoundSecAgg compared to the baselines of Section 3.3 in terms of the key metrics of Section 3.2. Next, we implement convolutional neural networks (CNNs) for image classification with MNIST [21], CIFAR-10, and CIFAR-100 [20] to investigate how the key metrics affect the test accuracy.

**Setup.** We consider a FL setting with $N = 120$ users, where the server aims to choose $K = 12$ users at every round. We study two settings for partitioning the CIFAR-100 dataset across the users.

- **IID Setting.** 50000 training samples are shuffled and partitioned uniformly across $N = 120$ users.
- **Non-IID Setting.** We distribute the dataset using a Dirichlet distribution [13], which samples $\mathbf{d}_c \sim \text{Dir}(\beta = 0.5)$ which specifying the prior class distribution over 100 classes, and allocate a portion $d_{c,i}$ of the class $c$ to user $i$. The parameter $\beta$ controls the heterogeneity of the distributions at each user, where $\beta \to \infty$ results in IID setting.

We implement a VGG-11 [29], which is sufficient for our needs, as our goal is to evaluate various schemes, not to achieve the best accuracy. The hyperparameters are provided in Appendix F.

**Modeling dropouts.** To model heterogeneous system, users have different dropout probability $p_i$ selected from $\{0.1, 0.2, 0.3, 0.4, 0.5\}$. At each round, user $i \in [N]$ drops with probability $p_i$.

**Implemented Schemes.** For the benchmarks, we implement the three baselines introduced in Sec. 3.3, referred to as *Random*, *Weighted Random*, and *Partition*. For Multi-RoundSecAgg, we construct three privacy-preserving families with different target multi-round privacy guarantees, $T = 6$, $T = 4$, and $T = 3$ which we refer to as Multi-RoundSecAgg ($T = 6$), Multi-RoundSecAgg ($T = 4$), and Multi-RoundSecAgg ($T = 3$), respectively. One can view the Random and Partition as extreme cases of Multi-RoundSecAgg with $T = 1$ and $T = K$, respectively. Table 1 summarizes the family size $R$ defined in Section 4.

| Scheme | Family size ($= R$) |
|---|---|
| Random selection | $\sim 10^{16}$ |
| Weighted random selection | $\sim 10^{16}$ |
| User partition | 10 |
| Multi-RoundSecAgg, T=6 | 190 |
| Multi-RoundSecAgg, T=4 | 4060 |
| Multi-RoundSecAgg, T=3 | 91389 |

**Table 1:** Family size with $N = 120$, $K = 12$.

**Key Metrics.** To numerically demonstrate the performance of the six schemes in terms of the key metrics defined in Sec. 3.2, at each round, we measure the following metrics.

- For the multi-round privacy guarantee, we measure the number of models in the partial sum that the server can reconstruct, which is given by $T^{(t)} := \min_{z \in \mathbb{R}^J} \|z^\top \mathbf{P}^{(t)}\|_0$, s.t. $\mathbf{P}^{(t)^\top} z \neq \mathbf{0}$. This corresponds to the weaker privacy definition of Remark 1. We use this weaker privacy definition as the random selection and the random weighted selection strategies provide the worst privacy guarantee even with this weaker definition, as demonstrated later. On the other hand, Multi-RoundSecAgg provides better privacy guarantees with both the strong and the weaker definitions.
- For the aggregation fairness gap, we measure the instantaneous fairness gap, $F^{(t)} := \max_{i \in [N]} F_i^{(t)} - \min_{i \in [N]} F_i^{(t)}$ where $F_i^{(t)} = \frac{1}{t+1} \sum_{l=0}^{t} \mathbb{1}\{\{p^{(l)}\}_i = 1\}$.
- We measure the instantaneous aggregation cardinality as $C^{(t)} := \frac{1}{t+1} \sum_{l=0}^{t} \|p^{(l)}\|_0$.

We demonstrate these key metrics in Figure 4. We make the following key observations.

- Multi-RoundSecAgg achieves better multi-round privacy guarantee than both the random selection and random weighted selection strategies, while user partitioning achieves the best multi-round privacy guarantee, $T = K = 12$. However, the partitioning strategy has the worst aggregation cardinality, which results in the lowest convergence rate as demonstrated later.
- Figure 5 demonstrates the trade-off between the multi-round privacy guarantee $T$ and the average aggregation cardinality $C$. Interestingly, Multi-RoundSecAgg when $T = 3$ or $T = 4$ achieves better multi-round privacy guarantee than both the random selection and the weighted random selection strategies while achieving almost the same average aggregation cardinality.

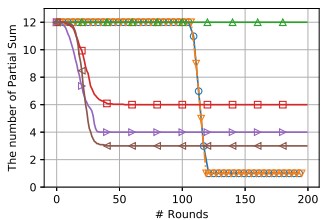
(a) Multi-round privacy guarantee.

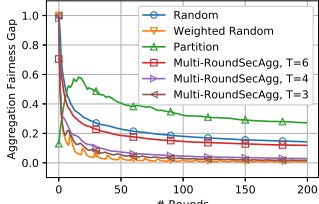
(b) Aggregation fairness gap.

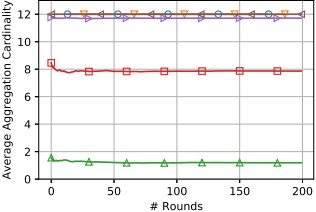
(c) Average aggregation cardinality.

**Figure 4:** The key metrics with $N = 120$ (number of users), $K = 12$ (number of selected users at each round).

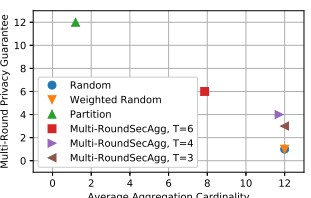
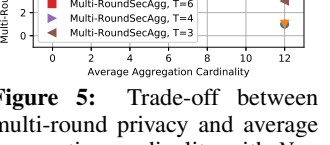

**Figure 5:** Trade-off between multi-round privacy and average aggregation cardinality with $N = 120$, $K = 12$.

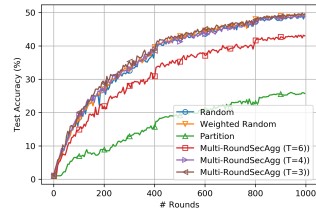
(a) IID data distribution.

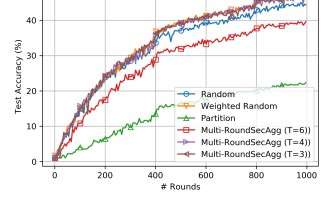
(b) Non-IID data distribution.

**Figure 6:** Training rounds versus test accuracy of VGG11 in [29] on the CIFAR-100 with $N = 120$ and $K = 12$.

**Remark 11.** (Multi-round Privacy of Random and Weighted Random). The multi-round privacy guarantees of Random and Weighted Random drop sharply as shown in Fig. 4(a) as the participating matrix $\mathbf{P}^{(t)} \in \{0, 1\}^{t \times N}$ becomes full rank with high probability when $t \geq N$, and hence the server can reconstruct the individual models by utilizing a pseudo inversion of the matrix $\mathbf{P}^{(t)}$. More precisely, Theorem 3 in Appendix H shows this *thresholding phenomenon*, where the probability that the server can reconstruct individual models after certain number of rounds converges to 1 exponentially fast.

**Key Metrics versus Test Accuracy.** To investigate how the key metrics affect the test accuracy, we measure the test accuracy of the six schemes in the two settings, the IID and the non-IID settings. Our results are demonstrated in Figure 6. We make the following key observations.

- In the IID setting, the Multi-RoundSecAgg schemes show test accuracies that are comparable to the random selection and random weighted selection schemes while the Multi-RoundSecAgg schemes provide higher levels of privacy. Specifically, the Multi-RoundSecAgg schemes achieve $T = 3, 4, 6$ based on the privacy-preserving family design while the random selection and random weighted selection schemes have $T = 1$, i.e., the server can learn an individual local model.
- In the non-IID setting, Multi-RoundSecAgg not only outperforms the random selection scheme but also achieves a smaller aggregation fairness gap as demonstrated in Fig. 4(b).
- In both IID and non-IID settings, the user partitioning scheme has the worst accuracy as its average aggregation cardinality is much smaller than the other schemes as demonstrated in Fig. 4(c).

We also implement additional experiments on MNIST and CIFAR-10 datasets in Appendix E and present ablation study for various settings of $(N, K, T)$ in Appendix G

# 7 Conclusion

Partial user participation may breach user privacy in federated learning, even if secure aggregation is employed at every training round. To address this challenge, we introduced the notion of long-term privacy, which ensures that the privacy of individual models are protected over all training rounds. We developed Multi-RoundSecAgg, a structured user selection strategy that guarantees long-term privacy while taking into account the fairness in user selection and average number of participating users, and showed that Multi-RoundSecAgg provides a trade-off between long-term privacy and average number of participating users (hence the convergence rate). Our experiments on the CIFAR-100, CIFAR-10, and MNIST datasets on both the IID and non-IID settings show that Multi-RoundSecAgg achieves comparable accuracy to the random selection strategy (which does not ensure long-term privacy), while ensuring long-term privacy guarantees.

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
