# OpenReview forum: "Securing Secure Aggregation: Mitigating Multi-Round Privacy Leakage in Federated Learning"
_NeurIPS.cc/2022/Conference — NeurIPS 2022 Submitted_

### Official Review · Reviewer_Rb6v · 2022-07-04

**Rating:** 6
**Confidence:** 4
**Soundness:** 3 good
**Presentation:** 2 fair
**Contribution:** 3 good

**Summary:**

The paper concentrates on privacy-leakage in federated learning over multiple rounds. The authors note that secure aggregation protocols only have guarantees for a single aggregation round, whereas in federated learning the model is trained iteratively.  The paper proposes to measure and mitigate this privacy leakage over iterations. The main idea in the mitigation is to choose fixed groups of parties for model updates, so that an adversary can only ever see a contribution from the fixed group. This makes identifying any single individual party's contribution based on differences between iterations harder, since the individual contribution is always mixed in with the other group members. The proposed method essentially involves creating fixed subsets of users who are always chosen together for model updates. Another consideration is fairness in the sens of each party contributing an equal number of updates: the authors also propose a method by which all parties are chosen for updates an equal number of times on expectation. The paper then analyses convergence properties of the proposed method, and tests it empirically with several standard network structures and datasets.

**Questions:**

1) In the experiments, is there some reason to stop training when you do? Would e.g. the partition baseline eventually reach ~ same performance, and the main issue is convergence speed, or is there some other reason why it would perform worse?

2) It seems that the proposed method assumes that the other parties in the same group are honest(?). Please mention the actual assumption as part of the the threat model.

3) I think Remark 7 on lines 290-295 would make more sense as a separate theorem: this seems like an important property that should be stated more formally.


#### Minor details:

Please add a table on the notation to Appendix: trying to follow the derivations can be frankly painful when one needs to hunt for the meaning of symbols all over the place. Fix typos on lines 113,224.

**Limitations:**

There is no discussion on the limitations of the proposed method, including in Section 7, which is pointed out as containing such discussion in the paper checklist.

**Strengths And Weaknesses:**

### Update after discussions

After the rebuttal and the discussions, I raise my score to recommend acceptance. I trust that the authors will improve the paper based on the discussions.

### Strengths

+ The idea of having some secure aggregation-type guarantees over several iterations seems like a fine one.
+ The theoretical results backing up the method are comforting, e.g., in the sense that the proposed method can be shown to converge under some fairly standard assumptions.


### Weaknesses

- The most glaring issue with the paper is that it completely omits discussing or even mentioning differential privacy (DP) and related definitions, which have been the standard privacy definition in machine learning for years, and that have also been widely used with federated learning (see e.g. Kairouz et al. 2019 for a quick overview of the basic idea in the context of federated learning, Dwork & Roth 2014 for a basic introduction to DP). Instead, the privacy discussed in the current paper is more related to secure computation, where the formal guarantees are on revealing only the end result of a computation. Calling such definitions privacy preserving is problematic for various reasons. One major issue is that the end result might well give out the input data in the clear without having any problems in satisfying this kind of definition of privacy-preserving (e.g. by choosing local loss functions s.t. the optimal value is an actual data point for a single client and 0 for everyone else would give out the single data point while still satisfying this kind of privacy guarantees).
The guarantees are also brittle in the sense that any side information might lead to a catastrophic privacy breach, where one or more input data points are suddenly revealed (e.g. consider what happens to others if one of the fixed group members gives out it's updates, or an adversary gains information on such values from other sources).
As an immediate fix, I would suggest including some discussion on the privacy-definition that also considers DP, clarifying the claims that this is the first paper that considers privacy-leakage in federated learning over iterations (e.g. lines 51, 84-88), and changing the emphasis to be less on general privacy-preservation and more specifically on what this method tries to protect. More interesting, although also more work-intensive and hence probably out of the question for now, would be to try and combine the current papers' results with DP and show that the proposed method does indeed enable better privacy by limiting the amount of information an adversary can gain.
- Related to the previous point: I have a hard time understanding what are the consequences of providing privacy in the sense used in this paper.


### References

Dwork & Roth 2014: The algorithmic foundations of differential privacy.
Kairouz et al. 2019: Advances and open problems in federated learning.

---

> ### Author Response · Authors · 2022-07-31
> **Response to Reviewer Rb6v**
>
> We thank the the reviewer for careful reading of our paper and for providing us helpful comments and suggestions. We have revised our manuscript to address the comments and provide our responses to your comments as follow.
>
> 1. Differential Privacy
> > As the reviewer pointed out, there are two lines of research to ensure privacy in FL (secure aggregation (SA) and differential privacy (DP)).\
> We consider the first line, SA, which utilizes secure multi-party computation to ensure that the server can learn nothing about the  local models in the information-theoretic sense beyond their aggregate. Our multi-round privacy definition ($T$) extends the privacy guarantee of SA from one round to all rounds by requiring that the server cannot learn the aggregate model of less than $T$ users. For the example of weakness of SA (the case that a single client sends local update while others send $0$) that the reviewer pointed out, we note that we consider the honest-but-curious user and server model, and this scenario is not allowed in this model. Hence, this is beyond the scope of the paper. Indeed, extending our results to the Byzantine model would be interesting future direction. We can prevent in such a scenario by combining SA with verification steps as in (So, et al., 2020).\
> In the second privacy-preserving technique, DP, each user adds artificial noises to the local model. In DP, however, the privacy guarantee sacrifices the model performance (privacy-utility trade-off). That is, SA ensures information-theoretic privacy without sacrificing accuracy, whereas DP ensures stronger privacy guarantee at the expense of the accuracy. As the reviewer pointed out, these two schemes (SA and DP) are complementary, i.e., all the benefits of DP can be applied to our approach by adding noise to the local models (Bonawitz, Kallista, et al, 2021). Indeed, combining our proposed scheme with DP would be an interesting future direction. We included this discussion in Section 2 of the revised paper. \
> We also modified our claims such that DP can be one of potential solutions for the multi-round privacy guarantee and our work is the first secure aggregation protocol to protect the long-term privacy in Section 1 of the revised paper.
>
> 2. Accuracy plot in the experiment
> > One of objectives in our experiments is to investigate how key metrics (especially, aggregation cardinality) affect the convergence speed. The accuracy plot is sufficient to show that the partition baseline has the worst convergence speed as it has the lowest aggregation cardinality while it will eventually reach the same performance.
> 3. Threat model
> > Yes, we assume that all clients and the server are honest-but-curious. We clarified this in the revised version of the paper.
> 4. Remark 7: non-linear reconstructions of aggregated models
> > As the reviewer suggested, we will write the Remark 7 as a Theorem in the final version of the paper.
> 5. Minor details
> > Thanks for pointing these out. We added the table on notation list to the modified Appendix and corrected the typos.
>
> \
> **References**
> * Bonawitz, Kallista, et al. "Federated Learning and Privacy: Building privacy-preserving systems for machine learning and data science on decentralized data." Queue 19.5 (2021): 87-114.
> * So, Jinhyun, Başak Güler, and A. Salman Avestimehr. "Byzantine-resilient secure federated learning." IEEE Journal on Selected Areas in Communications 39.7 (2020): 2168-2181.

---

> > ### Comment · Reviewer_Rb6v · 2022-08-03
> > **Some further clarifications**
> >
> > Thank you for the helpful comments.
> >
> > > In DP, however, the privacy guarantee sacrifices the model performance (privacy-utility trade-off)
> >
> > If one wants to protect user privacy, some form of randomization is required, which inevitably leads to a privacy-utility trade-off. Secure aggregation does not require such a trade-off, since it does not try to guarantee any user privacy, but protects against information leakage from the intermediate computations. This is exactly the point I would like to see clarified also in the paper.
> >
> > > we assume that all clients and the server are honest-but-curious
> >
> > Looking at it quickly, it would seem that assuming all honest-but-curious clients can leak the private models to other clients (for a simple example, take N=T=2, since the updated model from the server is just the sum of the local models, both clients can remove their own contribution to reveal the other clients' model). Is this intended behaviour, or did I misunderstand? If so, it would be good to note this in the paper.

---

> > > ### Author Response · Authors · 2022-08-04
> > > **Response to Reviewer Rb5v (Further clarifications)**
> > >
> > > Thanks for this note. We hope to address your concerns in detail below.
> > >
> > > 1. If one wants to protect user privacy, some form of randomization is required, which inevitably leads to a privacy-utility trade-off. Secure aggregation does not require such a trade-off, since it does not try to guarantee any user privacy, but protects against information leakage from the intermediate computations. This is exactly the point I would like to see clarified also in the paper.
> > > > Secure aggregation ensures information-theoretic privacy (Please see Theorem 1 and Appendix B in So et al, 2022 and Elkordy et al, 2022). In secure aggregation also there is a randomization as each user adds a mask to its local model before sharing it with the sever. However, the masks have this property that they cancel out at the server. Hence, there's no loss in accuracy when secure aggregation is used. DP methods ensure stronger privacy guarantee, however, the masks do not cancel with each other in such methods which leads to sacrificing the model performance. We will add a remark explaining this in the paper.
> > >
> > > 2. Looking at it quickly, it would seem that assuming all honest-but-curious clients can leak the private models to other clients (for a simple example, take N=T=2, since the updated model from the server is just the sum of the local models, both clients can remove their own contribution to reveal the other clients' model). Is this intended behaviour, or did I misunderstand? If so, it would be good to note this in the paper.
> > > > We guarantee that the server can at most reconstruct the sum of $T$ local models (from the aggregated models in multiple rounds). It is true that each user can remove his/her model and learn the aggregate model of the remaining $T-1$ users (if that user is involved in the $T$-sum). For instance, in your example of $N=T=2$, this results in learning the model of the other user (since $T-1=1$). But, this happens only for such a small value of $T$ that is typically not of interest. As $T$ gets larger than two, then no local model of the users are not learned and the users can only learn the aggregate of $T-1$ other users (the server still only learns the aggregate model of $T$ users only). We will add a remark to clarify this in the paper.
> > >
> > > **References**
> > > * So, Jinhyun, Corey J. Nolet, Chien-Sheng Yang, Songze Li, Qian Yu, Ramy E Ali, Basak Guler, and Salman Avestimehr. "Lightsecagg: a lightweight and versatile design for secure aggregation in federated learning." Proceedings of Machine Learning and Systems 4 (2022): 694-720.
> > > * Elkordy et al., "How Much Privacy Does Federated Learning with Secure Aggregation Guarantee?", in Proceedings on Privacy Enhancing Technologies (PoPETs), 2022.

---

> > > > ### Comment · Reviewer_Rb6v · 2022-08-09
> > > > **Thank you for the clarifications**
> > > >
> > > > I have raised my score to reflect this.

---

### Official Review · Reviewer_2oqH · 2022-07-13

**Rating:** 7
**Confidence:** 3
**Soundness:** 4 excellent
**Presentation:** 4 excellent
**Contribution:** 3 good

**Summary:**

Secure aggregation-based protocols can leak privacy over multiple training rounds in FL and do not provide long-term privacy guarantees. The partial user selection enables the reconstruction of the client's models. The authors propose a user-selection strategy that will provide long-term privacy guarantees and provide theoretical and convergence analysis. New metrics are proposed to capture the privacy guarantees.

**Questions:**

1- Will multi-secagg work for cross-silo setup with partial user selection?
2- Will experiments with N=K show any deviation?

minor edits:
Line 113: "their."
Line 41-42: Refer/Cite the basis
Line 224: "number of selected at each"

**Limitations:**

Limitations are discussed in form of assumptions and remarks.

**Strengths And Weaknesses:**

Strengths:
- The paper is well-written and clear.
- Multi-RoundSecAgg is a novel scheme that provides privacy in FL in long term. It also ensures a fair selection of clients and shows tradeoffs between convergence and privacy.
- Proposed metrics are interesting, they allow for comparison between different user selection schemes in terms of privacy and that all clients participate in an equal number of rounds.
- Comparison with baseline experiments demonstrates the effectiveness of the metrics proposed.
- The remarks and assumptions provided are a plus.

Weaknesses:
- The privacy leakage of the aggregated models over multiple rounds is not addressed. This may be considered as future work.
- Applicability to cross-device only is missing.

---

> ### Author Response · Authors · 2022-07-31
> **Response to Reviewer 2oqH**
>
> We thank the the reviewer for careful reading of our paper and for providing us helpful comments and suggestions. We have revised our manuscript to address the comments and provide our responses to your comments as follow.
>
> 1. The privacy leakage of the aggregated models over multiple rounds
> > Thanks for raising this point. Yes, investigating the privacy leakage from the aggregated models would be one of the interesting future direction.
> 2. Applicability to cross-device and cross-silo
> > Our proposed scheme can be applied to both cross-device and cross-silo settings. For instance, the large-scale cross-device FL system (whose population size is up to hundreds of millions) can utilize a hierarchical aggregating architecture to efficiently handle the massive communication and computational overhead (please see Figure 3 in Bonawitz et. al. 2019). This hierarchical architecture consists of \emph{Master Aggregator} and set of {Intermediate Aggregators}. Each intermediate aggregator aggregates hundreds or a thousand of clients and sends the aggregate results to the master aggregator. Hence, a small subset of the users (not all users) is aggregated by each intermediate aggregator, and our proposed algorithm can be applied to protect the long-term privacy for the intermediate aggregators.
> 3. Experiments with $N=K$
> > Thanks for your question. $K$ is less than $N$ in practical FL systems due to dropouts and the unavailability of some of the users or communication and computational efficiency (McMahan et al. 2017, Bonawitz et al. 2017). However, we will add this experiment to the final version of the paper for completeness.
> 4. Minor edits
> > Thanks for pointing these out, we modified the paper accordingly.
>
> \
> **References**
> * Bonawitz, Keith, et al. "Towards federated learning at scale: System design." Proceedings of Machine Learning and Systems 1 (2019): 374-388.
> * McMahan, Brendan, et al. "Communication-efficient learning of deep networks from decentralized data." Artificial intelligence and statistics. PMLR, 2017.
> * Bonawitz, Keith, et al. "Practical secure aggregation for privacy-preserving machine learning." proceedings of the 2017 ACM SIGSAC Conference on Computer and Communications Security. 2017.

---

### Official Review · Reviewer_Y96F · 2022-07-21

**Rating:** 8
**Confidence:** 3
**Soundness:** 4 excellent
**Presentation:** 4 excellent
**Contribution:** 4 excellent

**Summary:**

The paper explores the question of multi-round privacy leakage in federated learning with secure aggregation. To this end, the paper proposes a novel metric for multi-round privacy in FL. Existing FL secure aggregation protocols guarantee that the server can only learn the sum of N local models in any single round. The proposed multi-round privacy metric extends this guarantee to all training rounds. The paper motivates the need for multi-round privacy guarantees by demonstrating theoretically and experimentally that, under the assumption of partial user participation, existing secure aggregation protocols are vulnerable to attacks that aggregate models across multiple training rounds. To address this vulnerability, the paper proposes Multi-RoundSecAgg, a novel algorithm that provides a multi-round privacy guarantee. Theoretical analyses of the multi-round privacy, aggregation fairness gap and average aggregation cardinality guarantees of the proposed method are presented, as well as a convergence analysis. Finally, the proposed method is validated experimentally over range of FL settings and image datasets.

**Questions:**

H. Given that T=4 is not a very strong privacy guarantee and that for T=6 the convergence rate already starts to take a significant hit, I wonder whether you're not better off adding noise to the gradients as in DP-FedAvg to protect the local data. Did you consider this option?


**Limitations:**

The authors adequately address the limitations and potential negative societal impact of their work.

**Strengths And Weaknesses:**

# Strengths

A. The writing and figures are very clear.

B. Strong motivation. The paper demonstrates that, under the assumption of partial user participation, existing secure aggregation protocols are vulnerable to attacks that aggregate models across multiple training rounds. This is an important limitation of existing methods, which merits attention from the community.

C. Strong novelty. The paper presents a novel multi-round attack on existing secure aggregation protocols. The paper proposes a novel metric for multi-round privacy in FL. The paper proposes a novel secure aggregation method that provides a multi-round privacy guarantee. The paper provides novel theoretical results for both existing secure aggregation methods and the proposed method. All of these are important contributions.

D. Strong theoretical results. The paper provides the theoretical guarantees of the proposed Multi-RoundSecAgg method in terms of multi-round privacy, aggregation fairness gap and average aggregation cardinality. The paper also provides a convergence analysis of the proposed Multi-RoundSecAgg method. These results provide important insights about the proposed method and its functioning.

E. Strong experimental results. Relevant baselines are compared against over a range of FL settings and datasets. Across all evaluations, the proposed method shows a better multi-round privacy guarantee, yet comparable test accuracy compared to the baselines.


 # Weaknesses

F. Experimental evaluations with real-world datasets and SOA model architectures would strengthen the paper. That said, this is not a major issue given the novelty of the work and breadth of theoretical results.

G. Given the focus on reconstruction attacks, the paper is missing a discussion of differentially private federated learning as a potential solution to the multi-round privacy vulnerability.

---

> ### Author Response · Authors · 2022-07-31
> **Response to Reviewer Y96F**
>
> We thank the the reviewer for careful reading of our paper and for providing us helpful comments and suggestions. We have revised our manuscript to address the comments and provide our responses to your comments as follow.
>
> 1. Experimental evaluations with real-world dataset and SOTA models
> > Thanks for raising this point. We will include this in the final version of the paper.
> 2. Discussion of differential privacy and DP-FedAvg
> > We agree that differential privacy (DP) is one potential solution to the multi-round privacy leakage. In this paper, we focus on secure aggregation as DP sacrifices the model performance (privacy-utility trade-off). As the reviewer pointed out, these two schemes are complementary, i.e., all the benefits of DP can be applied to our approach by adding noise to the local models. Combining the proposed scheme with DP would be an interesting future direction. We included the discussion on the DP in Section 2 of the revised paper.

---

### Official Review · Reviewer_moUb · 2022-07-21

**Rating:** 4
**Confidence:** 3
**Soundness:** 3 good
**Presentation:** 3 good
**Contribution:** 1 poor

**Summary:**

The paper proposes a framework for sampling clients in each round of a Federated Learning training session. The proposed framework balances a notion of privacy (essentially a lower bound on number of sumands of any partial sum a server my reconstruct via linear combination of model updates retrieved throughout training), fairness gap (average gap in client participation across all rounds), and aggregation cardinality (average number of sumands in each round). The paper studies the relationship between these metrics, in relation to the convergence of the training. This is done by proposing new client sampling strategies, studying their properties, and comparing with naural baselines. The authors also provide experiments to support the analytical findings.

**Questions:**

- I am unsure about the practical relevance of the assumptions regarding values of K and N. For example, See Table 1 in https://arxiv.org/pdf/1912.04977.pdf. This seems to suggest that in practice N is huge with respect to K (several orders of magnitude), and that the number of training rounds is also smaller than N. I am thinking: K in the 100s, N in the millions, and number of training rounds L in the thousands. With this in my opinion more relevant setting in mind, a given client will participate in no more than 2 sums. The results in appendix H, as refelected in the experiments section, seems to rely on N and L being of the same order. It be useful for the paper to state upfront some examples of values of N, K, L of relevance to the paper, and found in practice.
- The privacy guarantee is defined with respect to a particular time of reconstruction attack. In fact the sort of differentiation attacks that you describe motivates differential privacy. Using DP, one could ensure that a particular sumand does not alter the sum too much. Of course, if a user participates in many rounds this guarantee will degrade quickly, but that goes back to the choice of N, K.
- How does one implement your proposed mechanim at the scale of the above numbers (and the ones in the referenced paper)? Are there any challenges there? The experiments use small values of K and N.

**Limitations:**

-

**Strengths And Weaknesses:**

- I am unsure about the practical relevance of the assumptions regarding values of K and N.
- There does not seem to exist a lot of literature in client selection in FL with the purpose of improving privacy with respect to the server. The framework of differential privacy seeems important in that discussion, but the submission does not discuss that.
- The privacy guarantee is defined with respect to a particular time of reconstruction attack.
- The technical results give a good intuition on how the different metrics interact, even with baseline client selection strategies, which might be useful to practitioners. However, the assumption that dropout probability is the same across all clients seems unrealistic (although hard to avoid).

---

> ### Author Response · Authors · 2022-07-31
> **Response to Reviewer moUb**
>
> We thank the the reviewer for careful reading of our paper and for providing us helpful comments and suggestions. We have revised our manuscript to address the comments and provide our responses to your comments as follow.
>
>
> 1. Differential Privacy
> > There are two lines of research to ensure privacy in federated learning (FL), secure aggregation (SA) and differential privacy (DP), and these two lines are complementary.\
> We consider the SA in our work, which utilizes secure multi-party computation to ensure that the server can learn nothing about the local models in the information-theoretic sense beyond their aggregate. In SA, each user adds a mask such that the masks of all users cancel out each other at the server and the server only learns the aggregate model. Our multi-round privacy definition ($T$) extends the privacy guarantee of SA from one round to all rounds by requiring that the server cannot learn the aggregate model of less than $T$ users.\
> The second technique is DP, in which each user adds artificial noises to the local models. These noises do not cancel out each other, unlike SA. Hence, in DP, the privacy guarantee sacrifices the model performance (privacy-utility trade-off). That is, SA ensures information-theoretic privacy without sacrificing accuracy, whereas DP ensures stronger privacy guarantee at the expense of the accuracy. It is worth noting that these two schemes are complementary, i.e., all the benefits of DP can be applied to our approach by adding noise to the local models (Bonawitz et al., 2021). Our goal is to understand the limitations of SA by itself, and combining SA with DP would be an interesting future direction. We included this discussion in Section 2 of the revised paper.
>
> 2. Practical relevance of the assumption regarding the values of $K, N, L$ and implementation of the proposed scheme at large-scale
> >Even in the large-scale FL, random selection with existing secure aggregations can reveal the information of individual models and our proposed scheme can be applied to the large-scale FL to protect the long-term privacy.
> It is true that $N$ and $L$ has the same order to enable the server to reconstruct the local models, but we would like to point out that even with very large $N$ it is also likely that participating matrix of subset of the users can be full rank due to the cyclic property of participation in real-world FL (Eichner et al., 2019). That is, during certain iterations, only small portion of users (e.g., small country with specific time zone) participate in the FL and hence the server can reconstruct the local models.
>     In addition, the real-world large-scale FL system (whose population size is up to hundreds of millions) may utilize a hierarchical aggregating architecture (multi-tenant FL) to efficiently handle the massive communication and computational overhead (please see Figure 3 in Bonawitz et. al. 2019). This hierarchical architecture consists of \emph{Master Aggregator} and set of {Intermediate Aggregators}. Each intermediate aggregator aggregates hundreds or thousand clients and sends the results to the master aggregator. Hence, $N$ in such a setting refers to only to the size of a  small subset of the users (not all users) that is aggregated by each intermediate aggregator. In this case, our proposed algorithm can be applied to protect the long-term privacy for the intermediate aggregators.
>
> 3. Same dropout probability
> >We do not assume all users have the same dropout probability throughout the paper, this is just in Theorem 1 to ensure the unbiased selection of users and simplifies the convergence guarantee.
> In our experiments, however, the dropout rates are selected among $\{0.1, 0.2, 0.3, 0.4, 0.5\}$ to model the system heterogeneity of the users and we empirically demonstrated that the proposed scheme works properly with heterogeneous dropout probability.
>
> \
> \
> **References**
> * Bonawitz, Kallista, et al. "Federated Learning and Privacy: Building privacy-preserving systems for machine learning and data science on decentralized data." Queue 19.5 (2021): 87-114.
> * Eichner, Hubert, et al. "Semi-cyclic stochastic gradient descent." International Conference on Machine Learning. PMLR, 2019.
> * Bonawitz, Keith, et al. "Towards federated learning at scale: System design." Proceedings of Machine Learning and Systems 1 (2019): 374-388.

---

> > ### Author Response · Authors · 2022-08-08
> > **Request for feedback and possible support (Message to Reviewer moUB)**
> >
> > Dear Reviewer moUB,
> >
> > We remain hopeful about the fact that the manuscript has drawn important praise from you as you mentioned in strengths. As now we believe we have addressed your concerns (thanks to your critical reviewing), we remain positive to receive your very important support. Moreover, we are really glad that reviewers found our problem of great relevance (Y96F, 2oqH, Rb6v), emphasized strong novelty of the proposed scheme (Y96F, 2oqH), believed our proposed three key metrics to be interesting and intuitive (Y96F, 2oqH), found our empirical/theoretical analysis useful for understanding how the key metrics interact in both existing and proposed FL schemes (moUb, Y96F, 2oqH, Rb6v), and stated that the paper is well written and well presented (Y96F, 2oqH).
> >
> > As we are closing towards the discussion, please feel free to reach us out if you need any further clarification on any of the topics.
> >
> > Thank you
> >
> > Paper11033 Authors

---

> > > ### Author Response · Authors · 2022-08-09
> > > **Authors' final day remark: feedback request on our rebuttal and engage in discussion**
> > >
> > > Dear reviewer moUb,
> > >
> > > As you know today is the last day of our discussion, meaning last opportunity to have us help you with any doubt, clarification. As this paper has drawn significant praise, and now three accepts with one updated score(4->6), we would really like to know if we can receive your support in acceptance and accordingly updated score. *We have taken each of your comments and doubts in to our serious account to write the detailed rebuttal, and thus hope they will satisfy you and help you update the score*. If you still feel you need help in any of the part, please reach us out.
> > >
> > > Thank you
> > >
> > > Paper11033 Authors

---

### Author Response · Authors · 2022-08-04
**Message to all reviewers**

We would like to thank you again for the time you dedicated to reviewing our paper. We believe that we have addressed your concerns. Since the end of discussion period is getting close and we have not heard back from you yet, we would appreciate it if you kindly let us know of any other concerns you may have, and if we can be of any further assistance in clarifying any other issues.

---

### Meta-Review · Area_Chair_Jszt · 2022-08-28

**Recommendation:** Reject
**Confidence:** Less certain

**Metareview:**

The secure aggregation protocol is a basic building block in federated learning. It allows a central entity to learn the sum of vectors held by the clients, without being able to (noticeably) distinguish two different sets of vectors that lead to the same sum. This paper studies the question of how much privacy is leaked across multiple rounds of the secure aggregation protocol. It proposes a new “multi-round privacy” notion which aims to quantify the privacy leakage across multiple rounds of the secure aggregation protocol. The paper shows that the commonly used random subsampling protocol would violate this notion of privacy after a number of rounds that is linear in the number of users. The paper gives another algorithm, Multi-RoundSecAgg, which has better privacy properties, according to the proposed notion.

There are several drawbacks of this paper, several pointed out by the reviewers:

1) It is well-known that federated learning without differential privacy (DP) is vulnerable to the leakage of user’s sensitive information (e.g., inversion attacks), which this paper seeks to protect. E.g., from the aggregate vectors, one could directly read sensitive information pertaining to one user (e.g., if the individual vectors are sparse and have disjoint supports).
The argument provided by the authors in the rebuttal -- that they chose not to consider differential privacy since it hurts model accuracy -- is not satisfactory. In fact, the paper itself argues that even for the proposed multi round notion of privacy, for a more stringent privacy level to give the same utility, a larger number of rounds is needed until convergence. So even with the new notion, privacy does not come for free and has to be traded-off at least against efficiency (more precisely convergence speed).
It is also worth pointing out that in the case of DP, the multi-round leakage is already handled through composition theorems, and accounted for. See, e.g., the “Practical and Private (Deep) Learning Without Sampling or Shuffling” paper of Kairouz et al. (ICML 2019).

2) As pointed out by one of the reviewers, reconstructing an individual model after a number of rounds larger than the number of users is not realistic. Typical federated learning models are indeed trained for a number of rounds much smaller than the number of users. A detailed discussion of this limitation would be expected in the paper.

3) The claim that the worst-case is achieved when the local models do not change at all seems a bit hand-wavy to me. On a high level, it seems to ignore the correlation between the different models that would arise during the training process. Due to this correlation, it might be that all the users’ local models converge to the first user’s model. In this case, wouldn’t that be worse than having the models not change at all? Formalizing this claim, and its proof, would be helpful.

4) Given that the paper proposes a new privacy notion, discussing the qualitative properties and limitations of this notion is expected. E.g., does this notion compose across multiple applications? How does this notion behave in the presence of side information?

Overall, in its present form, and for the reasons listed above, the paper falls short of the acceptance bar at NeurIPS.

**Award:**

No

---

### Decision · Program_Chairs · 2022-09-14

Reject